# A Tale of Two Seasons: Distinct Seasonal Viral Communities in a Thermokarst Lake

**DOI:** 10.3390/microorganisms11020428

**Published:** 2023-02-08

**Authors:** Valérie Langlois, Catherine Girard, Warwick F. Vincent, Alexander I. Culley

**Affiliations:** 1Département de Biochimie, de Microbiologie et de Bio-Informatique, Université Laval, Québec, QC G1V 0A6, Canada; 2Centre D’études Nordiques (CEN), Université Laval, Québec, QC G1V 0A6, Canada; 3Institut de Biologie Intégrative et des Systèmes (IBIS), Université Laval, Québec, QC G1V 0A6, Canada; 4Takuvik International Research Laboratory, Université Laval, Québec, QC G1V 0A6, Canada; 5Département des Sciences Fondamentales, Université du Québec à Chicoutimi, Chicoutimi, QC G7H 2B1, Canada; 6Département de Biologie, Université Laval, Québec, QC G1V 0A6, Canada

**Keywords:** viral ecology, thermokarst lake, metagenomics, subarctic

## Abstract

Thermokarst lakes are important features of subarctic landscapes and are a substantial source of greenhouse gases, although the extent of gas produced varies seasonally. Microbial communities are responsible for the production of methane and CO_2_ but the “top down” forces that influence microbial dynamics (i.e., grazers and viruses) and how they vary temporally within these lakes are still poorly understood. The aim of this study was to examine viral diversity over time to elucidate the seasonal structure of the viral communities in thermokarst lakes. We produced virus-enriched metagenomes from a subarctic peatland thermokarst lake in the summer and winter over three years. The vast majority of vOTUs assigned to viral families belonged to Caudovirales (Caudoviricetes), notably the morphological groups myovirus, siphovirus and podovirus. We identified two distinct communities: a dynamic, seasonal community in the oxygenated surface layer during the summer and a stable community found in the anoxic water layer at the bottom of the lake in summer and throughout much of the water column in winter. Comparison with other permafrost and northern lake metagenomes highlighted the distinct composition of viral communities in this permafrost thaw lake ecosystem.

## 1. Introduction

Northern environments are among the most affected by climate change, with temperature increases that are up to nearly four times greater than the global average warming rate [1]. The Great Whale River region on the eastern coast of Hudson Bay (Nunavik, Canada) has been subjected to a substantial increase in air temperature over the past few decades [2]. The result is a major change in the landscape through reduced snow and ice cover [3], thawing permafrost [4] and the growth of thermokarst ponds and lakes [5,6]. 

When ice-rich permafrost thaws, the ground weakens and collapses, leaving a hole that fills with meltwater and rainwater to form a thermokarst lake, a common freshwater feature of the arctic and subarctic region [7]. These permafrost thaw lakes have been identified as biogeochemical hotspots in the landscape for the degradation of complex organic carbon molecules and the production of methane and carbon dioxide [8]. Since thermokarst lakes are small and shallow, their water quickly becomes carbon rich [9] from various allochthonous terrestrial sources. These include leaching from thawing permafrost near the lakes [10] and the decomposition of organic matter in the surrounding peatland [9]. This high carbon intake increases microbial activity and therefore CO_2_ concentration [11], and creates anoxic conditions at the bottom of the lakes that promote methanogenesis [12]. During winter, which accounts for most of the year in these regions (e.g., 8 months in northern Quebec), the water columns within the ice-covered lakes are rapidly depleted of oxygen by respiration [13] because the ice cover acts as a barrier for atmospheric gas exchanges and sunlight penetration. Microbes (bacteria, archaea, protists, fungi, and associated viruses) are the most abundant forms of life in thermokarst lakes and control the net emission of methane. Understanding the factors that influence these communities is essential to ultimately understand the contribution of thermokarst lakes to greenhouse gas production. Microbial communities are influenced by “bottom up” environmental conditions such as light, temperature and nutrient availability, among other factors. However, they are also influenced by “top down” factors such as grazing and viral activity. Although viruses have proven to be important in other northern environments [14] (e.g., Yau et al., 2018), their impact on microbes in thermokarst lakes and on seasonal scales in particular, remains essentially unknown.

Viruses are the most abundant and diverse biological entities in aquatic environments [15] and have been characterized in both freshwater [16,17,18] and saltwater systems [19,20,21,22] as well as in various soil environments [23,24] including permafrost [25,26]. They can affect the food web, microbial population equilibrium [21,27,28,29] and even biogeochemical cycles [29,30]. There have been few studies on viral ecology in the North [14], and little is known about the impact of viruses on Arctic and subarctic ecosystems or their temporal dynamics in these warming environments. 

Previous work on the thermokarst valleys near Hudson Bay has shown that bacterial community diversity varies across valleys [31] and seasons [32,33]. These studies demonstrated that the thermokarst lakes contained microbial groups similar to those found in other freshwater environments, but that they are enriched with methanogenic and methanotrophic organisms. Chloroviruses and myoviruses that infect photosynthetic organisms such as cyanobacteria and chlorophytes, along with their host communities, have been studied by Lévesque et al. [34] in several lakes and ponds in the region. Their study showed that both host and viral communities varied significantly between lake types and illustrated how the variation of virus community diversity correlated with changes in their host counterparts. However, this study focused on two specific groups of viruses and omitted most viral groups that cannot be detected through PCR analyses. Additionally, a metagenomic study of viruses from the cellular fraction (>0.22 µm) of the water column across seasons was conducted by Girard et al. [35] and showed a clear difference in viral diversity at the lake surface between the summer and the winter. Their study was limited to the upper part of the lake and focused specifically on viruses associated with the cellular size fraction.

Our study aimed to encompass the diversity of winter and summer extracellular viral communities in a thermokarst lake based on metagenomics. We anticipated that the thermokarst lakes would have two distinct seasonal communities. This is based on the observed water conditions and previous studies that suggest that oxygen and temperature are important drivers of viral and microbial diversity [32,33,35]. We sampled the lake on three occasions over the course of three years: once during late summer, once during winter and again in late summer. Two distinct viral communities were detected. The distributions of these communities were related to oxygen and temperature, with one community present in the warm surface waters during the summer. The other was confined to the cold bottom waters during the summer but occupied much of the water column during the winter. Our data suggest that there is little exchange between the water layers. A comparison of our data with other subarctic and boreal aquatic environments showed minimal overlap, suggesting that thermokarst viruses form distinct communities. The clear shifts in viral diversity suggest that viruses have an active role that changes over time.

## 2. Materials and Methods

### 2.1. Study Site and Sampling

Our study site was the Sasapimakwananistikw River Valley (SAS), a peatland located in Nunavik, Canada, at the northern limit of the sporadic permafrost zone (Figure 1A). The vegetation is dominated by *Carex*, *Eriophorum*, *Comarum palustre*, *Sphagnum* and various bryophytes, and is dotted with palsas and thermokarst lakes. Located in the subarctic area, this region undergoes long, cold winters (8 months) and short, warm summers (4 months). During the winter, the landscape is covered with a thick layer of snow and ice, daylight lasts for as little as 9 h per day [36] and the temperature can reach a low of −38 °C [37]. During the summer, the maximum number of daylight hours per day is 19 [36] and air temperatures can reach as high as 28 °C [37]. 

Samples were collected in triplicate from a small dystrophic thermokarst lake, SAS2A (55°13.160′ N; 77°41.806′ W) in late August 2015, early March 2016 and early September 2017. The samples were collected at the surface (0 m; by hand) and oxycline (0.5 m; mid-oxycline, with a 2 L Van Dorn sampling bottle) in 2015, under the ice (0.5 m; with a 2 L Limnos sampling bottle) in 2016, and at the surface (0.5 m; just above the oxycline, with a 2 L Limnos sampling bottle) and from the bottom (2.5 m; with a 2 L Limnos sampling bottle) in 2017. Limnological proprieties of the lake (temperature, conductivity, dissolved oxygen (DO) and pH) were measured with a YSI^™^ 600R profiler in 2015 and 2016 [39] and a conductivity temperature depth (C.T.D.) profiler RBR XR620 in 2017. The samples were kept in the dark in a cooler for up to 4 h until processing in the lab. Further details on sample collection are given in Vigneron et al. [32].

### 2.2. DNA Collection, Extraction and Sequencing

The samples were not pre-concentrated but were prefiltered through 0.22 μm Sterivex™ filters and then onto 0.02 μm Anotop™ filters using sterile syringes (2017) or a peristaltic pump (2015 and 2016). The volume of sample filtered onto the Anotop™ ranged from 10 to 50 mL and varied with particle loading. The filters were immediately frozen and stored at −70 °C or below until nucleic acid extraction. Total nucleic acids were extracted from the Anotop filters using the method described in Mueller et al. (2014) [40] and then purified with the Power Clean Pro DNA Clean up kit (QIAGEN, Hilden, Germany) to remove potential inhibitors. Metagenomic libraries were then prepared using the Accel NGS®1S (Swift Biosciences, Ann Arbor, MI, USA) kit after concentrating with the DNA Clean & Concentrator (Zymo Research, Irvine, CA, USA) kit as recommended by the manufacturer. The DNA was sequenced on an Illumina HiSeq at the Génome Québec facility (McGill University) (2 × 125 bp). 

### 2.3. Sequence Analysis and Identification of Viral Contigs

The read quality was examined with FastQC v0.11.2 [41]. Raw reads were then trimmed using Trimmomatic v0.36 [42] to remove Illumina TruSeq adapter sequences and low-quality nucleotides (phred score under 20). Small (under 30 bp) and unpaired reads were discarded. A single de novo co-assembly was created using MegaHit v1.1.1 [43] with kmer sizes ranging from 27 to 99 and the --kmin-1pass option recommended for large metagenomes. This co-assembly was made from all 15 libraries described earlier, two prior HiSeq libraries made from the first replicate of both surface and oxycline waters from 2015, and one MiSeq library made from the second replicate of oxycline water from 2015. The resulting assembly was evaluated with quast v4.6.1 [44]; the results can be found in Appendix A. Viral contigs were identified using VirSorter v.1.0.3 [45] and VirFinder v1.1 [46]. VirSorter was used with the virome decontamination option and VirFinder was trained using a model that included eukaryotic viruses (https://github.com/jessieren/VirFinder; accessed on 2 November 2018). Contigs of 10kb or more belonging to VirSorter categories 1 and 2 and/or with a VirFinder confidence score of 90% or higher were kept for further analysis. From the 5,445,783 contigs of the total assembly, 5637 viral contigs were retained.

### 2.4. Abundance Table and Statistical Analysis

Trimmed reads from each library were mapped to the viral contigs with Bowtie2 v2.3.4.1 [47] and SAMtools v1.8 [48] and the alignment files were piped through the program Read2Ref Mapper v1.1.0 [25] to produce a normalized abundance table. The read counts for each viral operational taxonomic unit (vOTU) in each library were first normalized to contig size and then to library size in base pairs. The vOTU table only contained contigs with a coverage of 85% or more and accounts for reads mapping on 90% of their length with 95% identity. The 5% of the least abundant contigs overall were removed from the table as well as contigs that did not appear in at least two of the triplicates for at least one sampling condition. After processing the data, the table contained the abundances for 2265 viral contigs representing vOTUs. The values in this table were used for our alpha diversity analysis. The table was then transformed with the Hellinger method (decostand{vegan}) for distance-based analyses and a binary version of the table was used to determine presence or absence. Taxonomy was assigned up to the family level with the VPF-Class tool v1.0 [49]. Only assignments with both a confidence score and membership that were >20% were considered for this analysis. Assignments to a viral family with a membership ratio >50% were kept as is and vOTUs that showed assignments to multiple families from a single order with membership ratios between 20% and 50% were assigned to the order level. The vOTUs that did not meet the criteria were considered unassigned. Although the classification used by VPF-Class is no longer accepted by the international committee on taxonomy of viruses (ICTV) as described by Turner et al. [50], the former classification was retained to offer more opportunities for comparisons with other studies.

Most statistical analyses were conducted in R v3.6.2. Alpha diversity was assessed by the number of contigs found in each sample, Shannon’s richness index and Pielou’s evenness index (diversity{vegan}; v2.5-6). A Venn diagram was created using the binary table data and the R package {venn} (v1.9). A Euclidean distance matrix was calculated from the Hellinger-transformed abundance table and used to produce a neighbor-joining tree (nj{ape} + root{ape}; v5.3) and an ordination, which was visualized with a PCoA (ordinate and plot_ordination{phyloseq}; Bioconductor v3.10). A random forest analysis was conducted with QIIME v1.9.1 [51] using the “leave-one-out” cross-validation option for small datasets to identify the most important vOTUs for differentiating viral communities between sampling conditions. This classification algorithm identifies the features (vOTUs) most important to determine the group identity of a sample (sampling condition) by creating multiple decision trees based on bootstrapping and random feature selection. After creating the trees, the algorithm then predicts the group identity of samples. Trees made with features that are most useful for differentiating between given categories give better predictions than trees made from less important features. The 50 vOTUs that were most indicative of the differences between sampling conditions were considered bio-indicators of the different sampling conditions.

### 2.5. Comparison with Relevant Published Datasets 

The viral communities examined in this study were compared with the putative intracellular viral communities from the oxycline samples from 2015 and winter samples from 2016, as described in Girard et al. [35]. For this analysis, trimmed reads from Girard et al. were added to our own dataset and a new abundance table was produced, treated and analyzed following the same method as described previously. The results analyzed by PCoA were visualized with an ordination plot. We also used DRAM-V (v1.2.4) [52] to search for integrase genes in the vOTU sequences. 

Finally, we compared the viral communities from this study with those of other aquatic and permafrost communities described in Appendix A. Briefly, we compared our data with libraries produced from triplicate samples from Lake Simoncouche and Lake Croche [53], two lakes in the boreal zone of southern Québec; two water samples from Lough Neagh [16], a lake in Ireland; four soil samples from Bonanza Creek [12], a peat valley in Alaska; and 257 samples from the Stordalen Mire [54], a peat valley in Sweden. All external libraries were quality checked and trimmed following the same method as previously described. For libraries from Lough Neagh, sequences were removed when the mean quality over 10 bp in length reached 24 or lower. Samples from all lakes and from Stordalen Mire were assembled individually whereas libraries from Bonanza Creek were pooled by soil type prior to assembly. To limit technical biases when comparing our libraries with external libraries, triplicates from our study were pooled and reassembled before being processed in the same way as the published libraries. De novo assemblies were made with MegaHit v1.1.1 [43] (lakes, Bonanza Creek, thermokarst lake) or with v.1.2.9 through the KBase server (Stordalen Mire). Viral contigs were identified using VirSorter v1.0.3 [45] (lakes, Bonanza Creek, thermokarst lake) or v.1.0.5 through the KBase server (Stordalen Mire). Contigs from the Stordalen Mire were then pooled by sample type (permafrost, fen, bog). The reads were merged using BBmerge before being mapped to the viral contigs from the corresponding assemblies with Bowtie2 v2.3.4.1 [47] (lakes, Bonanza Creek, thermokarst lake) or v2.3.5.1 (Stordalen Mire). Mapped reads were extracted and used as inputs for a similarity analysis using Libra v1.2.8 [55]. Outputs from Libra were visualized using R and Cytoscape v3.8.0 [56]. 

## 3. Results and Discussion

### 3.1. Site Description

The water conditions in the SAS2A lake were highly variable between seasons and with depth. Cold, anoxic water was found at the bottom of the lake in summer and under the ice in winter. Warmer, more oxygenated water was found at the surface during the summer (Figure 1). These conditions are consistent with data collected in previous studies [39,57] that show the same pattern over multiple years. 

The SAS2A lake is dark in color due to a high concentration of colored dissolved organic matter (13.7 mg DOC L^−1^). The summer water column is slightly acidic (pH 6) and stratified with an upper layer that is warmer (12.75 °C) and more oxygenated (47% O_2_) than the bottom layer (6.08 °C and 0% O_2_) [34]. The thickness of the surface layer varies due to partial mixing of the water column by wind. Similar conditions were observed in 2017 (Figure 1B). This lake is known to be a CH_4_ and CO_2_ emitter [57,58]. During summer, CH_4_ concentrations are higher at the bottom of the lake with values ranging from 2.5 μM to 300 μM. During the winter, the lake is covered with snow and ice. Water temperatures range from 0 °C at the surface to 3.5 °C at the bottom. The water is more acidic (pH 5) and completely anoxic [34], with methane and DOC concentrations of approximately 200 μM and 18.3 mg L^−1^, respectively. General sample characteristics can be found in Appendix A. 

### 3.2. Community Diversity

We found between 167 and 977 vOTUs per library (Figure 2), with two outliers in 2015 (surface replicate 2 and oxycline replicate 1) (Appendix A). The Shannon index for the viral community was high across all samples and showed no significant difference between the communities (*p* > 0.05). This is consistent with results from previous studies that also found no significant difference in the Shannon index for viral [34,35] or bacterial [31] communities between years, seasons or depths from the SAS2A lake. [34] All of the communities also had high Pielou’s evenness indices, with the summer surface community in 2017 being significantly less even than the summer surface community in 2015, the winter surface community in 2016, and the summer bottom community in 2017. All of the communities also had high Pielou’s evenness indices. However, the summer surface community in 2017 was significantly less even compared to the surface communities in summer 2015 and winter 2016 and the summer bottom community in 2017 (*p* < 0.05). This lower value for evenness could be the result of a few dominant vOTUs in the community that were found in larger abundance. These vOTUs most likely originate from a cellular microbial bloom resulting from favorable light and oxygen conditions at the surface. It is possible that the very high evenness and Shannon index of diversity scores indicate that the lakes harbor some of the most diverse aquatic viral communities yet to be observed. Another explanation is that these values are the result of biases in the vOTU assignment process.

In our samples, 1376 vOTUs (60.7%) were assigned to viral families and 42 vOTUs (1.9%) were assigned to the order *Caudovirales* with VPF-Class. Of the vOTUs assigned to viral families, the vast majority belonged to *Caudovirales* families with 837 *Myoviridae*, 276 *Siphoviridae* and 185 *Podoviridae*. Fewer vOTUs were assigned to *Lavidaviridae* (13), *Tectiviridae* (3), *Phycodnaviridae* (2), *Nudiviridae* (1), *Iridoviridae* (1) and *Mimiviridae* (1). The other 904 vOTUs (39.9%) were left unassigned. It should be noted that in the most recent classification system for the taxonomy of bacteriophages, the order Caudovirales has been replaced by the class Caudoviricetes (all tailed prokaryotic viruses with icosahedral capsids and double-stranded DNA), and that the morphologically based families (here considered ‘morphofamilies’) Myoviridae, Podoviridae and Siphoviridae have been abolished [50]. It is notable that of the 50 vOTUs most indicative of the sampling conditions (depth and season) of each community (Figure 3, all the taxonomically assigned vOTUs were identified as members of viral families that infect prokaryotes both under the former and the new taxonomy. This is consistent with the cellular community composition that is dominated by archaeal and bacterial taxa (including cyanobacteria), with only a few eukaryotes [31,32,34,59,60].

### 3.3. Contrasting Summer Surface Viral Communities

Few vOTUs were shared among all sampling conditions (Figure 4A), which illustrates the striking differences between the viral communities by season and by depth within the lake. Of the 2265 vOTUs, only 2.52% (57) were found in every library. Of these ubiquitous vOTUs, six were some of the most important vOTUs for identifying the sampling conditions during sample collection using the random forest analysis (Figure 3). Of these six, half were found in greater abundance in the oxygenic conditions and the rest showed the opposite trend, being more abundant in all anoxic sampling conditions and less abundant in oxygenic waters. Three out of the six vOTUs were unclassified, while the other three were classified as *Myoviridae,* S*iphoviridae* and *Tectiviridae*.

The surface and oxycline communities from 2015 were indistinguishable from one another according to the PCoA and neighbor-joining tree (Figure 4B,C). This is also supported by the large number of shared vOTUs that are unique to these two groups (247–10.91%) and the very low number of vOTUs that are unique to one depth and not to the other (Figure 4A). This suggests that the samples from the oxycline and the surface have similar viral communities, despite the difference in O_2_ concentrations. These results correspond with those from Lévesque et al. [34], where the diversity of bacteria, protists and select viral groups showed no difference between depths. The uniformity of the viral community throughout the surface and the oxycline in lake SAS2A might be the result of the proximity of the oxycline to the surface of the lake (Figure 1B). Despite differences in oxygen concentrations, the fact that both layers are in the photic zone [61,62] may have resulted in microbial communities being dominated by phototrophs and their viruses. Another contributing factor to the similarity of the communities could be the influence of rain squalls, wind gusts and water inputs, among others. These events could cause the upper layer of the lake to mix on a more regular basis, resulting in indistinguishable microbial communities even after temperature, conductivity and oxygen gradients have been re-established.

The summer upper water column (surface and oxycline) communities appear to differ from year to year. The 2015 and 2017 summer upper water column communities cluster separately on the PCoA plot (Figure 4C). This distinction is likely driven by the 450 (19.27%) vOTUs that are unique to the surface in 2017 and the 257 (10.91%) vOTUs unique to the upper water column in 2015 (Figure 4A). None of these vOTUs were present in the libraries from the anoxic samples. Additionally, 18 of the 50 vOTUs that were determined by the random forest analysis to be the most indicative of the different sampling conditions (Figure 3) were only found in the summer upper water column. These results indicate that a substantial part of the viral community changes annually, which is consistent with data from other aquatic environments [16,63,64]. The notable absence from the 2016 winter library of viruses that were found in both the 2015 and 2017 libraries suggests an annual reintroduction of summer vOTUs from seeding sources other than the bottom layer. The annual reseeding of viruses from allochthonous sources is consistent with previously observed exchanges between the lakes and the peatland communities in the SAS2 valley [65]. Alternatively, this difference between summer surface communities could be linked to normal intra-seasonal variability [16,64] related to small changes in physicochemical conditions and/or blooms of specific hosts.

We found a high proportion of vOTUs that were unique to a single sampling time. In fact, most (59%) of the vOTUs identified in our study were only found in libraries from a single year (Figure 4A). These viruses most likely represent viral strains that did not persist from one year to the next due to a loss of fitness, as observed in other systems [66,67]. Alternatively, these findings could simply have been a product of the limitations of our methods. It is conceivable that these vOTUs were present throughout the sampling period but their abundances were lower than the detection limits of our methods.

### 3.4. Deep Persistent Viral Communities

We found that the viral communities from anoxic sampling conditions (the winter samples from 2016 and the summer bottom samples from 2017) were closely related. They shared 316 (13.95%) vOTUs that were not found in any other samples (Figure 4A). The similarity between these two communities was also evident in the neighbor-joining tree and PCoA graph (Figure 4B,C). This suggests that the anoxic community was stable, persistent, and varied little from year to year, reflecting the fact that the SAS2A water column does not fully mix each year [13].

Though relatively similar to one another, the two anoxic communities remained distinct. This is in part because of the 293 (12.94%) and the 298 (13.16%) vOTUs unique to the 2017 summer bottom water and winter 2016 water column, respectively (Figure 4A). As discussed previously for the summer oxygenic community results, the difference between 2016 and 2017 anoxic communities could be the result of interannual variation in the physicochemistry of the lake, changes in the host community and/or differences in the factors responsible for viral decay. These differences could also be linked to a few season-specific vOTUs. Of the 50 most important vOTUs for identifying the sampling conditions, 9 were associated with only summer samples, including the anoxic bottom samples of 2017, and were absent from the winter samples (Figure 3). This seems to further indicate that the anoxic community found in SAS2A is persistent and inhabits much of the water column during the winter. It also suggests that the rare seeding of this community that does occur, takes place during the summer months, possibly through summer mixing events during open water conditions.

### 3.5. Viral Reservoirs

Our data suggest that lysogeny is not a dominant overwintering strategy used by upper water column viruses. We hypothesized that some of the summer-only vOTUs might be temperate phages that persisted as prophages throughout the winter. In other aquatic environments, lysogeny has been shown to be a viable strategy for viruses to persist through winter months when the conditions are less favorable for bacterial growth [68,69]. We investigated this hypothesis by comparing our data with the viral communities identified by Girard et al. [35] in the intracellular fraction from the same lake. A PCoA analysis showed that communities from the free virus fraction in our study and the intracellular fraction from Girard et al. [35] were not distinct (Appendix A). This suggests that the intracellular viruses identified by Girard et al. were primarily lytic (i.e., producing virions) and not lysogens. Additionally, we did not observe an enrichment in vOTUs with integrases in samples from the summer upper water column relative to the winter (Appendix A). Our data therefore suggest that either summer upper water column viruses are present in abundances that are below the detection limits of our method, or that they are in fact absent from the lake in the winter, both as free viruses and/or prophages. 

We propose that the seasonal dynamics of viral communities in these thermokarst lakes might be explained by the following scenario (Figure 5). The lakes harbor persistent and seasonal viral communities. The persistent viral community, which is linked to cold anoxic water conditions in which methanogens dominate [32], occupies much of the water column during the winter and is pushed to the bottom of the lake in the spring as the summer oxygenic upper water layer forms. Because the viruses that form the summer surface community were not found in the winter samples and do not seem to use lysogeny as an over-wintering strategy, we hypothesize that these viruses and their hosts are reintroduced to the lake each summer from external sources in the surrounding watershed (Figure 5). These reservoirs might include soil systems, notably the eroding palsa mounds throughout the valley, as well as percolating water from the surrounding peatland, lake sediment, melting snow and ice and aerosols.

### 3.6. Divergent Thermokarst Lake Viral Populations

To better understand the relatedness of thermokarst viruses to other environments, we compared our data with five other systems: three lakes and two permafrost-associated peatlands. All data used in this comparison were based on shotgun metagenomes sequenced with Illumina technologies (MiSeq or HiSeq). We selected two North American boreal lakes (Lakes Croche and Simoncouche) [53] that are covered with ice in the winter, and a lake from Ireland (Lough Neagh) [16] that is ice-free all year round and for which summer and winter metagenomes were available. We also compared our samples with metagenomes from two permafrost-associated peatlands (Stordalen Mire and Bonanza Creek) [12,54], which shared features with the SAS valley peatland. We expected to see more overlap between the SAS2A and the lake viral communities than those of the peatland soils. This supposition was based on previous data that suggested that SAS2A archaeal communities were more similar to aquatic microbiota than those of soils [31].

Comparisons of viral populations using Libra [55] showed no resemblance between SAS viral communities and the other environments that were included in the analysis (Figure 6). While Simoncouche, Croche and Lough Neagh showed a limited detectable resemblance to one another (<3%), the permafrost systems were even less similar. According to our analysis, there was no overlap between the SAS2A viral communities and the peatland permafrost soils or the lakes. However, Girard et al. [35] did find a few near-perfect matches between SAS2A viral sequences and other North American lake sequences via the IMG/VR database [70]. The lack of similarity between our metagenomes and those we compared them with show the distinct and divergent nature of the viral communities in thermokarst lakes. Together, these results indicate that although the SAS2A lake is a freshwater system and is in closely related to the permafrost soil systems that surround it, there appears to be very little exchange. These results correspond with results from previous studies that found that bacterial [31,59] and viral [34] communities were different across two hydrologically disconnected valleys.

## 4. Conclusions

This study characterized the viral communities of a thermokarst lake, which are important producers of greenhouse gases such as methane and carbon dioxide in the subarctic. We observed two distinct communities in our samples. The first community was found only in the oxygenic part of the water column in the summer, and the second community was found both under the ice in winter and at the bottom of the lake in summer. The summer oxygenic community appeared to be absent during the winter. It is possible that this viral community is reintroduced each year, leading to pronounced shifts in community composition between years. Conversely, the persistent community seems to inhabit the lake year-round with little seeding from the summer upper water column community. Based on these observations, we suggest that the persistent anoxic community occupies the entire water column during winter and recedes to the bottom of the lake in summer when the seasonal community is reintroduced at the surface from outside seeding sources. These findings would benefit from further spatial and temporal sampling to characterize the magnitude of stochastic variation among the lakes. Nonetheless, the large differences observed here give new insight into an understudied component of the microbial community that likely plays a role in the top-down control of the microbes involved in the production of greenhouse gases. 

## Figures and Tables

**Figure 1 microorganisms-11-00428-f001:**
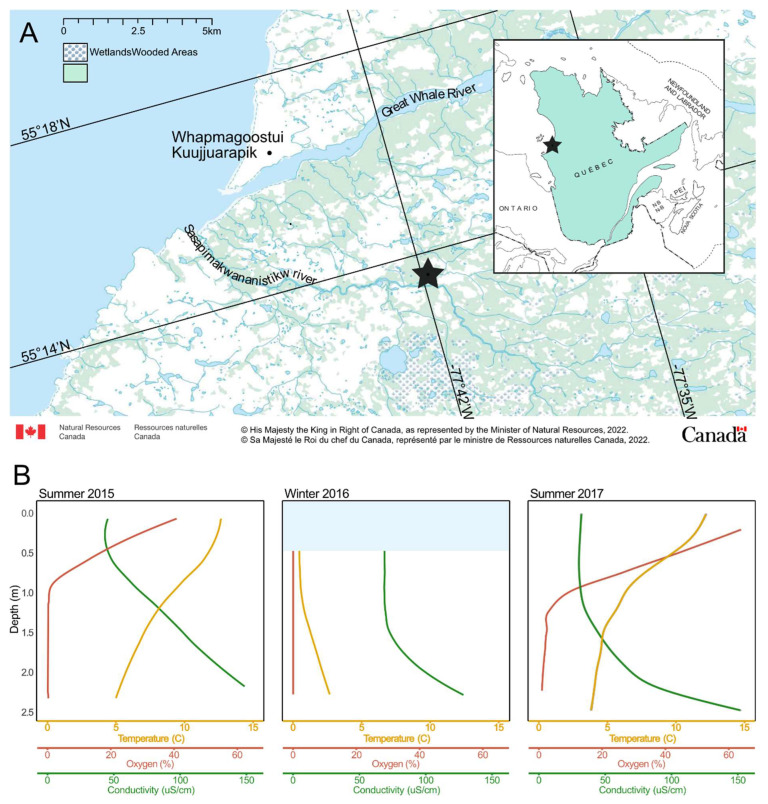
(**A**) Map of the study region, including the SAS valley, indicated by a star. The map was made with tools provided by the Atlas of Canada [38]. (**B**) Profiles for the three sampling dates. The light blue bar in the Winter 2016 panel indicates the presence of ice.

**Figure 2 microorganisms-11-00428-f002:**
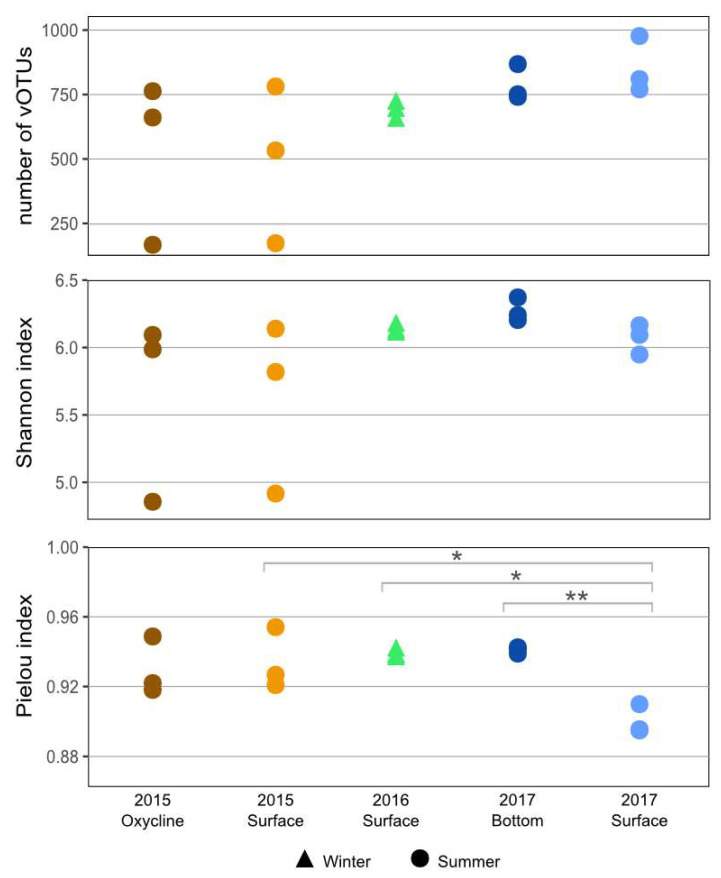
Viral sample richness in summer and winter surface samples from SAS2A lake. Diversity results are shown based on the number of viral taxa present (**upper** panel), the Shannon diversity index (**middle** panel) and the Pielou’s index (**lower** panel). Significant differences are indicated with * for *p* ≤ 0.05 and ** for *p* ≤ 0.01.

**Figure 3 microorganisms-11-00428-f003:**
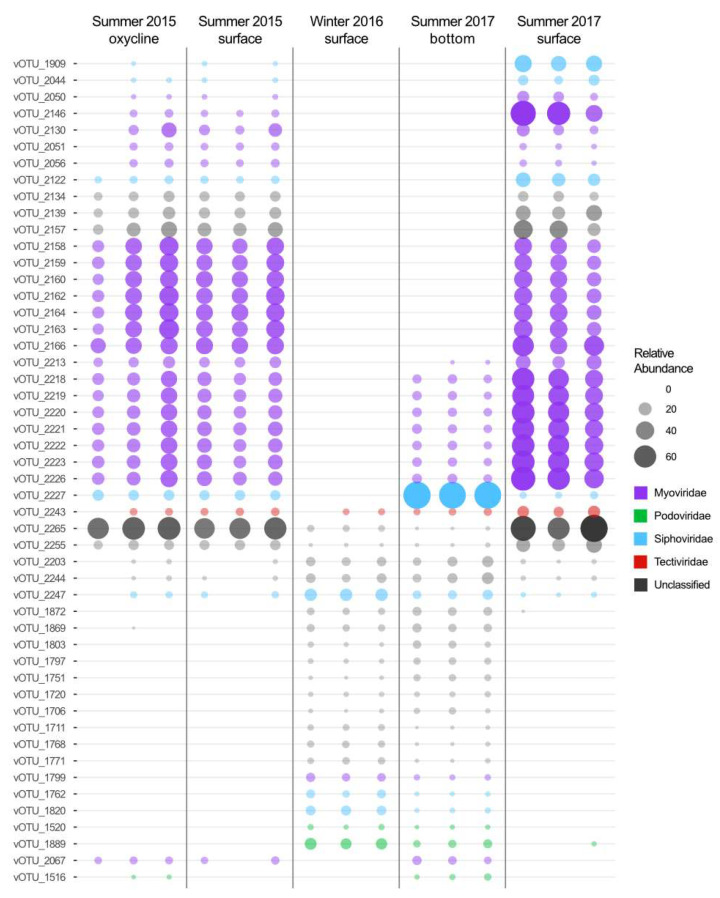
Relative abundance and assigned taxonomy of the 50 vOTUs most representative of the difference between sampling conditions. The 50 vOTUs were selected through random forest analysis with four sampling condition categories (summer surface, summer oxycline, summer bottom and winter). Taxonomy was assigned with VPF-Class.

**Figure 4 microorganisms-11-00428-f004:**
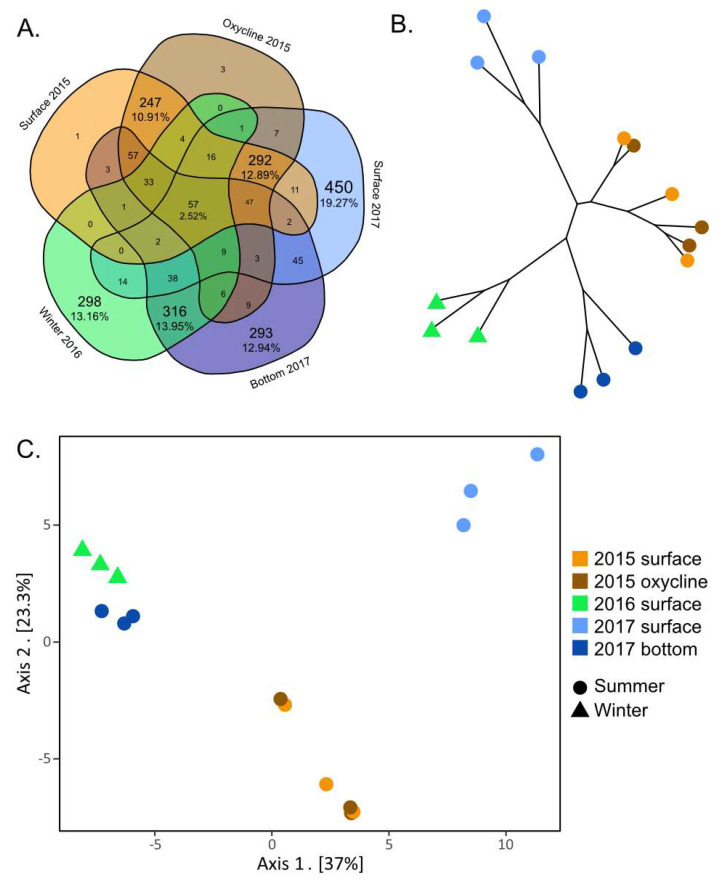
Representation of beta diversity of viral communities from the SAS2A lake. Beta diversity is represented using (**A**) a Venn diagram, (**B**) a neighbor-joining tree and (**C**) a principal coordinates analysis (PCoA) graph. The neighbor-joining tree and PCoA graph were both made using the Euclidean distance between viral communities.

**Figure 5 microorganisms-11-00428-f005:**
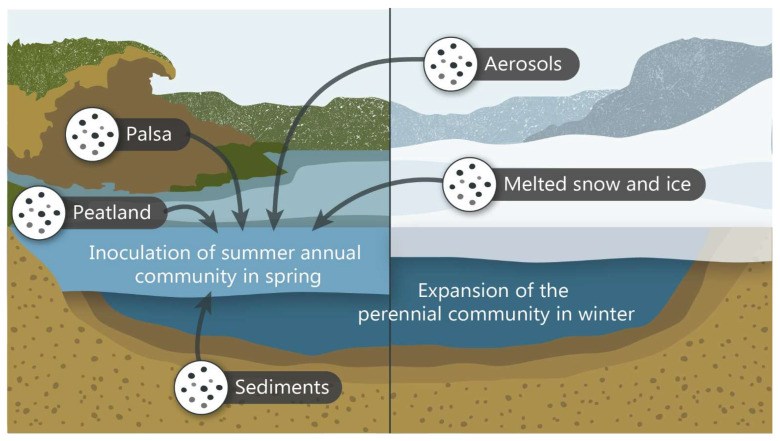
Visual representation of the proposed model of viral community transition in thermokarst lakes between summer and winter.

**Figure 6 microorganisms-11-00428-f006:**
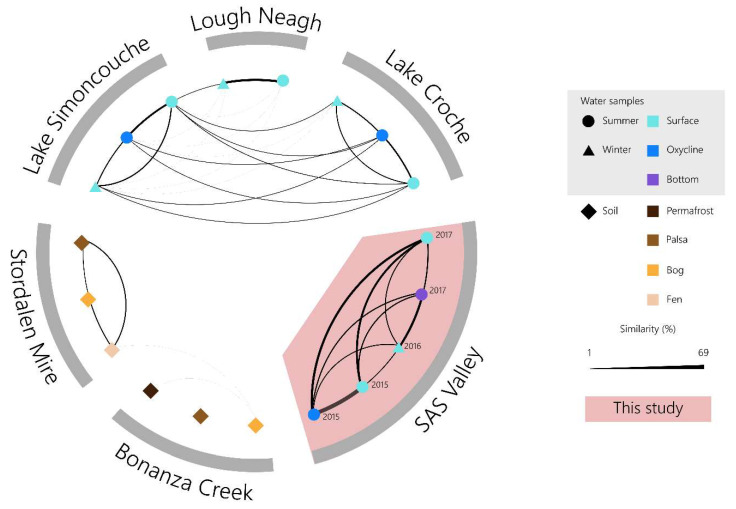
Genetic similarity between viral communities from various environments. Similarity was calculated with Libra using reads matched to contigs that were identified as viral (category 1 and 2) by VirSorter.

## Data Availability

The raw reads have been deposited in NCBI under project number PRJNA801668. The environmental data for the SAS2 lake is available in the Nordicana D48 archive at https://nordicana.cen.ulaval.ca/dpage.aspx?doi=45588CE-5A12A84DFAAF4D36 (accessed on 26 July 2019).

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
