# Peer review of "A Tale of Two Seasons: Distinct Seasonal Viral Communities in a Thermokarst Lake"

_microorganisms, 2023, doi:10.3390/microorganisms11020428_

Round 1

Reviewer 1 Report

The manuscript submitted by Langlois et al. (ID: 2165408) examine viral diversity over time in thermokarst lakes. Their study identified two distinct viral communities: a more dynamic community which prevailed during summer at the surface water and, a stable perennial one at the bottom and winter season. Overall the manuscript has been well written and the results are quite interesting. I don’t have any major issues with regard to its content and writing. I only have some minor comments to make.

L114-116: make it more clear. Is 0.5 m surface or oxycline? What water sampler was used for collection of samples? In what time interval were the samples transported from the collection site to laboratory for further processing?

L122: Were the water samples concentrated for the extraction of nucleic acid? How much of water samples were filtered for DNA extraction.

L125: reference for Meuller et al. 2014 needs to be added.

What is the trophic status of investigated lakes?

Author Response

The manuscript submitted by Langlois et al. (ID: 2165408) examine viral diversity over time in thermokarst lakes. Their study identified two distinct viral communities: a more dynamic community which prevailed during summer at the surface water and, a stable perennial one at the bottom and winter season. Overall the manuscript has been well written and the results are quite interesting. I don’t have any major issues with regard to its content and writing. I only have some minor comments to make.

Response: Thank you for this positive feedback, and for these constructive review comments.

L114-116: make it more clear. Is 0.5 m surface or oxycline?

Response: We have clarified this by noting ‘mid-oxycline’ (2015) and just above the oxycline (2017), which can also be seen in the figure.  

What water sampler was used for collection of samples?

Response: This is now specified for each sample depth and date.

In what time interval were the samples transported from the collection site to laboratory for further processing?

Response: This has been added to line 124.

L122: Were the water samples concentrated for the extraction of nucleic acid? How much of water samples were filtered for DNA extraction.

Response: No they were not concentrated, now specified. The volume filtered onto the sample was dependent on particle loading.

L125: reference for Meuller et al. 2014 needs to be added.

Response: Now added as reference 39.

What is the trophic status of investigated lakes?

Response: The lakes are dystrophic, now added.

Reviewer 2 Report

This study investigated the seasonal dynamics of viral communities in a subarctic Thermokarst lake. The authors found that the viral community structure in surface water were distinct between two seasons, while it was consistent in bottom water. This finding is of scientific interest. However, there are still some key issues that should be solved.

1. Since only lake was investigated, the title of this study should be more concentrated.

2. Line25-29,description of results are repeated. What is the essential conclusion of this study?

3. Line 244, add a period after 2017

4. Figure 3C/Figure S1, check if surface 2016 was collected at summer represented by triangle.

5. Line348-349, why the obvious difference between two anoxic communities can indicate their persistence? Why one depth (0.5m) can represent the “entire water column in winter”?

6. Line377-381, these two sentences are contradictory. Whether perennial viral community can be found in winter?

7. The conclusions of this study cannot be well supported by limited samples. Viral samples were collected only once in winter and bottom water in summer was collected only once. Probability of stochastic event cannot be excluded.

Author Response

This study investigated the seasonal dynamics of viral communities in a subarctic Thermokarst lake. The authors found that the viral community structure in surface water were distinct between two seasons, while it was consistent in bottom water. This finding is of scientific interest. However, there are still some key issues that should be solved.

Response: Thank you for this positive feedback, and for these constructive review comments.

  1. Since only lake was investigated, the title of this study should be more concentrated.

Response: Now changed to ‘a thermokarst lake’ as requested.

  1. Line25-29,description of results are repeated. What is the essential conclusion of this study?

Response: We have modified the sentences to clarify our conclusion about the two distinct communities (seasonal versus perennial) and have deleted the repetition. 

  1. Line 244, add a period after 2017

Response: Now added.

  1. Figure 3C/Figure S1, check if surface 2016 was collected at summer represented by triangle.

Response: Thank you for picking up this error, which we have now corrected.

  1. Line348-349, why the obvious difference between two anoxic communities can indicate their persistence? Why one depth (0.5m) can represent the “entire water column in winter”?

Response: These differences are very small relative to oxygenic versus anoxic, perhaps due to stochastic variation (see point 7). The profiles in winter do not show the sharp stratification as in summer, and anoxia occurs throughout the water column. However, we have modified this to ‘much of the water column’ to be more conservative.

  1. Line377-381, these two sentences are contradictory. Whether perennial viral community can be found in winter?

Response: This has been clarified to emphasize that it is the summer surface community that is not found in winter, while the anoxic water community persists throughout the year.  

  1. The conclusions of this study cannot be well supported by limited samples. Viral samples were collected only once in winter and bottom water in summer was collected only once. Probability of stochastic event cannot be excluded.

Response: We have now addressed these issues in the final section of the Conclusions by drawing attention to the need for further sampling, including to identify stochastic variations.

Reviewer 3 Report

I like this article very much. And in my opinion, is a very important contribution to aquatic virus knowledge. I have only a few suggestions or comments:

In some parts of the results and discussion, the authors declare vOTUs were unclassified. Can this fact be deduced as a possibility of new viral species?

Suggestion: position the figures closest to the text that describes them. This would facilitate the reader's analysis.

Line 268: "...dominated by archaeal and bacterial taxa, with only a few eukaryotes." Did cyanobacteria make up the lake's bacterial community?

The sentence posted in the abstract: Microbial communities are responsible for the production of methane and CO2.   Do the authors think that viruses can control microbial communities and influence methane production?

Line 90:  ... suggest that oxygen and temperature are important drivers of viral and microbial diversity. Does the temperature also influence virus communities?

Author Response

I like this article very much. And in my opinion, is a very important contribution to aquatic virus knowledge. I have only a few suggestions or comments:

Response: Thank you for this positive feedback, and for these constructive review comments.

In some parts of the results and discussion, the authors declare vOTUs were unclassified. Can this fact be deduced as a possibility of new viral species?

Response: This is a very interesting point. But given the extremely incomplete nature of the viral database we are reluctant to suggest this.

Suggestion: position the figures closest to the text that describes them. This would facilitate the reader's analysis.

Response: We are somewhat limited by the page format, but we have now repositioned figures.   

Line 268: "...dominated by archaeal and bacterial taxa, with only a few eukaryotes." Did cyanobacteria make up the lake's bacterial community?

Response: Cyanobacteria are a component of the bacterioplankton of the lake, and this is now noted, with new reference 60 inserted.

The sentence posted in the abstract: Microbial communities are responsible for the production of methane and CO2.   Do the authors think that viruses can control microbial communities and influence methane production?

Response: This is highly possible, but we have no direct evidence. However, we have now inserted a reference to the dominance of methanogens in the anoxic environments of our sampling site.

Line 90:  ... suggest that oxygen and temperature are important drivers of viral and microbial diversity. Does the temperature also influence virus communities?

Response: We have now modified this text to include this information.

Round 2

Reviewer 2 Report

Abstract: Background description is too long. Key results should be provided.

Line333: persistant?persistent

Line338-339: “This suggests that the bottom summer community is stable, persistent, and varies little from year to year”, use “anoxic community” instead of “bottom summer community”

Line381:”Because the viruses that form the summer community were not found in the winter samples”, the summer surface community

Author Response

Abstract: Background description is too long. Key results should be provided.

Response: We removed the last part of background description regarding impact of viruses in environments and added the results related to phylogeny. We also rephrased the last sentence to be more concise.

Line333: persistant?Persistent

Response: This was a french speaking author typo. Thanks for catching it, it has been corrected.

Line338-339: “This suggests that the bottom summer community is stable, persistent, and varies little from year to year”, use “anoxic community” instead of “bottom summer community”

Response: Now changed for anoxic community.

Line381:”Because the viruses that form the summer community were not found in the winter samples”, the summer surface community

Response: Now added.